# Linear Contextual Bandits with Knapsacks

**Shipra Agrawal**[*]                    **Nikhil R. Devanur**[†]

## Abstract

We consider the linear contextual bandit problem with resource consumption, in addition to reward generation. In each round, the outcome of pulling an arm is a reward as well as a vector of resource consumptions. The expected values of these outcomes depend linearly on the context of that arm. The budget/capacity constraints require that the total consumption doesn't exceed the budget for each resource. The objective is once again to maximize the total reward. This problem turns out to be a common generalization of classic linear contextual bandits (linContextual) [8, 11, 1], bandits with knapsacks (BwK) [3, 9], and the online stochastic packing problem (OSPP) [4, 14]. We present algorithms with near-optimal regret bounds for this problem. Our bounds compare favorably to results on the unstructured version of the problem [5, 10] where the relation between the contexts and the outcomes could be arbitrary, but the algorithm only competes against a fixed set of policies accessible through an optimization oracle. We combine techniques from the work on linContextual, BwK and OSPP in a nontrivial manner while also tackling new difficulties that are not present in any of these special cases.

## 1 Introduction

In the contextual bandit problem [8, 2], the decision maker observes a sequence of contexts (or features). In every round she needs to pull one out of $K$ arms, after observing the context for that round. The outcome of pulling an arm may be used along with the contexts to decide future arms. Contextual bandit problems have found many useful applications such as online recommendation systems, online advertising, and clinical trials, where the decision in every round needs to be customized to the features of the user being served. The *linear contextual bandit* problem [1, 8, 11] is a special case of the contextual bandit problem, where the outcome is linear in the feature vector encoding the context. As pointed by [2], contextual bandit problems represent a natural half-way point between supervised learning and reinforcement learning: the use of features to encode contexts and the models for the relation between these feature vectors and the outcome are often inherited from supervised learning, while managing the exploration-exploitation tradeoff is necessary to ensure good performance in reinforcement learning. The linear contextual bandit problem can thus be thought of as a midway between the linear regression model of supervised learning, and reinforcement learning.

Recently, there has been a significant interest in introducing multiple "global constraints" in the standard bandit setting [9, 3, 10, 5]. Such constraints are crucial for many important real-world applications. For example, in clinical trials, the treatment plans may be constrained by the total availability of medical facilities, drugs and other resources. In online advertising, there are budget constraints that restrict the number of times an ad is shown. Other applications include dynamic pricing, dynamic procurement, crowdsourcing, etc.; see [9, 3] for many such examples.

In this paper, we consider the **linear contextual bandit with knapsacks** (henceforth, linCBwK) problem. In this problem, the context vectors are generated i.i.d. in every round from some unknown distribution, and on picking an arm, a reward and *a consumption vector* is observed, which depend

---

[*]Columbia University. `sa3305@columbia.edu`.
[†]Microsoft Research. `nikdev@microsoft.com`.

linearly on the context vector. The aim of the decision maker is to maximize the total reward while ensuring that the total consumption of every resource remains within a given budget. Below, we give a more precise definition of this problem. We use the following notational convention throughout: vectors are denoted by bold face lower case letters, while matrices are denoted by regular face upper case letters. Other quantities such as sets, scalars, etc. may be of either case, but never bold faced. All vectors are column vectors, i.e., a vector in $n$ dimensions is treated as an $n \times 1$ matrix. The transpose of matrix $A$ is $A^\top$.

**Definition 1** (linCBwK). *There are $K$ "arms", which we identify with the set $[K]$. The algorithm is initially given as input a budget $B \in \mathbb{R}_+$. In every round $t$, the algorithm first observes context $\mathbf{x}_t(a) \in [0,1]^m$ for every arm $a$, and then chooses an arm $a_t \in [K]$, and finally observes a reward $r_t(a_t) \in [0,1]$ and a $d$-dimensional consumption vector $\mathbf{v}_t(a_t) \in [0,1]^d$. The algorithm has a "no-op" option, which is to pick none of the arms and get $0$ reward and $\mathbf{0}$ consumption. The goal of the algorithm is to pick arms such that the total reward $\sum_{t=1}^T r_t(a_t)$ is maximized, while ensuring that the total consumption does not exceed the budget, i.e., $\sum_t \mathbf{v}_t(a_t) \le B\mathbf{1}$.*

*We make the following stochastic assumption for context, reward, and consumption vectors. In every round $t$, the tuple $\{x_t(a), r_t(a), \mathbf{v}_t(a)\}_{a=1}^K$ is generated from an unknown distribution $\mathcal{D}$, independent of everything in previous rounds. Also, there exists an unknown vector $\mu_* \in [0,1]^m$ and a matrix $W_* \in [0,1]^{m \times d}$ such that for every arm $a$, given contexts $x_t(a)$, and history $H_{t-1}$ before time $t$,*

$$\mathbb{E}[r_t(a)|x_t(a), H_{t-1}] = \mu_*^\top x_t(a), \quad \mathbb{E}[\mathbf{v}_t(a)|x_t(a), H_{t-1}] = W_*^\top x_t(a). \tag{1}$$

*For succinctness, we will denote the tuple of contexts for $K$ arms at time $t$ as matrix $X_t \in [0,1]^{m \times K}$, with $\mathbf{x}_t(a)$ being the $a^{th}$ column of this matrix. Similarly, rewards and consumption vectors at time $t$ are represented as the vector $\mathbf{r}_t \in [0,1]^K$ and the matrix $V_t \in [0,1]^{d \times K}$ respectively.*

As we discuss later in the text, the assumption in equation (1) forms the primary distinction between our linear contextual bandit setting and the general contextual bandit setting considered in [5]. Exploiting this linearity assumption will allow us to generate regret bounds which do not depend on the number of arms $K$, rendering it to be especially useful when the number of arms is large. Some examples of this include recommendation systems with large number of products (e.g., retail products, travel packages, ad creatives, sponsored facebook posts). Another advantage over using the general contextual bandit setting of [5] is that we don't need an oracle access to a certain optimization problem, which in this case is required to solve an NP-Hard problem. (See Section 1.1 for a more detailed discussion.)

We compare the performance of an algorithm to that of an optimal adaptive policy that knows the distribution $\mathcal{D}$ and the parameters $(\mu_*, W_*)$, and can take into account the history up to that point, as well as the current context, to decide (possibly with randomization) which arm to pull at time $t$. However, it is easier to work with an upper bound on this, which is the optimal expected reward of a static policy that is required to satisfy the constraints only in expectation. This technique has been used in several related problems and is standard by now [14, 9].

**Definition 2** (Optimal Static Policy). *A context-dependent non-adaptive policy $\pi$ is a mapping from context space $[0,1]^{m \times K}$ to $\Omega = \{\mathbf{p} \in [0,1]^K : \|\mathbf{p}\|_1 \le 1\}$, where $\pi(X)_i$ denotes the probability of playing arm $i$ when the context is $X$, and $1 - \sum_{i=1}^K \pi(X)_i$ is the probability of no-op. Define $\mathbf{r}(\pi)$ and $\mathbf{v}(\pi)$ to be the expected reward and consumption vector of policy $\pi$, respectively, i.e.*

$$\mathbf{r}(\pi) \quad := \quad \mathbb{E}_{(X,\mathbf{r},V) \sim \mathcal{D}}[\mathbf{r}\pi(X)] = \mathbb{E}_{X \sim \mathcal{D}}[\mu_*^\top X \pi(X)]. \tag{2}$$

$$\mathbf{v}(\pi) \quad := \quad \mathbb{E}_{(X,\mathbf{r},V) \sim \mathcal{D}}[V\pi(X)] = \mathbb{E}_{X \sim \mathcal{D}}[W_*^\top X \pi(X)]. \tag{3}$$

$$Let \quad \pi^* \quad := \quad \arg\max_\pi \quad T\,\mathbf{r}(\pi) \quad such\ that \quad T\,\mathbf{v}(\pi) \le B\mathbf{1} \tag{4}$$

*be the optimal static policy. Note that since no-op is allowed, a feasible policy always exists. We denote the value of this optimal static policy by $OPT := T\,\mathbf{r}(\pi^*)$.*

The following lemma proves that OPT upper bounds the value of an optimal *adaptive* policy. Proof is in Appendix B in the supplement.

**Lemma 1.** *Let $\overline{OPT}$ denote the value of an optimal adaptive policy that knows the distribution $\mathcal{D}$ and parameters $\mu_*, W_*$. Then $OPT \ge \overline{OPT}$.*

**Definition 3** (Regret). *Let $a_t$ be the arm played at time $t$ by the algorithm. Then, regret is defined as*

$$regret(T) := OPT - \sum_{t=1}^{T} \mathbf{r}_t(a_t).$$

## 1.1 Main results

Our main result is an algorithm with near-optimal regret bound for linCBwK.

**Theorem 1.** *There is an algorithm for linCBwK such that if $B > m^{1/2}T^{3/4}$, then with probability at least $1 - \delta$,*

$$regret(T) = O\left((\tfrac{OPT}{B} + 1)m\sqrt{T\log(dT/\delta)\log(T)}\right).$$

**Relation to general contextual bandits.** There have been recent papers [5, 10] that solve problems similar to linCBwK but for general contextual bandits. In these papers the relation between contexts and outcome vectors is arbitrary and the algorithms compete with an arbitrary fixed set of context dependent policies $\Pi$ accessible via an optimization oracle, with regret bounds being $O\left((\tfrac{OPT}{B} + 1)\sqrt{KT\log(dT|\Pi|/\delta)}\right)$. These approaches could potentially be applied to the linear setting using a set $\Pi$ of linear context dependent policies. Comparing their bounds with ours, in our results, essentially a $\sqrt{K\log(|\Pi|)}$ factor is replaced by a factor of $m$. Most importantly, we have no dependence on $K$,[3] which enables us to consider problems with large action spaces.

Further, suppose that we want to use their result with the set of linear policies, i.e., policies of the form, for some fixed $\boldsymbol{\theta} \in \Re^m$,

$$\arg\max_{a \in [K]} \{\mathbf{x}_t(a)^{\top}\boldsymbol{\theta}\}.$$

Then, their algorithms would require access to an "Arg-Max Oracle" that can find the best such policy (maximizing total reward) for a given set of contexts and rewards (no resource consumption). In fact, by a reduction from the problem of learning halfspaces with noise [16], we can show that the optimization problem underlying such an "Arg-Max Oracle" problem is NP-Hard, making such an approach computationally expensive. The proof of this is in Appendix C in the supplement.

The only downside to our results is that we need the budget $B$ to be $\Omega(m^{1/2}T^{3/4})$. Getting similar bounds for budgets as small as $B = \Theta(m\sqrt{T})$ is an interesting open problem. (This also indicates that this is indeed a harder problem than all the special cases.)

**Near-optimality of regret bounds.** In [12], it was shown that for the linear contextual bandits problem, no online algorithm can achieve a regret bound better than $\Omega(m\sqrt{T})$. In fact, they prove this lower bound for linear contextual bandits with *static* contexts. Since that problem is a special case of the linCBwK problem with $d = 1$, this shows that the dependence on $m$ and $T$ in the above regret bound is optimal upto log factors. For general contextual bandits with resource constraints, the bounds of [5, 10] are near optimal.

**Relation to BwK [3] and OSPP [4].** It is easy to see that the linCBwK problem is a generalization of the linear contextual bandits problem [1, 8, 11]. There, the outcome is scalar and the goal is to simply maximize the sum of these. Remarkably, the linCBwK problem also turns out to be a common generalization of the bandits with knapsacks (BwK) problem considered in [9, 3], and the online stochastic packing problem (OSPP) studied by [13, 6, 15, 14, 4]. In both BwK and OSPP, the outcome of every round $t$ is a reward $r_t$ and a vector $\mathbf{v}_t$ and the goal of the algorithm is to maximize $\sum_{t=1}^{T} r_t$ while ensuring that $\sum_{t=1}^{T} \mathbf{v}_t \le B\mathbf{1}$. The problems differ in how these rewards and vectors are picked. In the OSPP problem, in every round $t$, the algorithm may pick any reward,vector pair from a given set $A_t$ of $d + 1$-dimensional vectors. The set $A_t$ is drawn i.i.d. from an unknown distribution over *sets of vectors*. This corresponds to the special case of linCBwK, where $m = d + 1$ and the context $\mathbf{x}_t(a)$ itself is equal to $(r_t(a), \mathbf{v}_t(a))$. In the BwK problem, there is a fixed set of arms, and for each arm there is an unknown distribution over reward,vector pairs. The algorithm picks an arm and a reward,vector pair is drawn from the corresponding distribution for that arm. This

corresponds to the special case of linCBwK, where $m = K$ and the context $X_t = I$, the identity matrix, for all $t$.

We use techniques from all three special cases: our algorithms follow the primal-dual paradigm and use an online learning algorithm to search the dual space, as was done in [3]. In order to deal with linear contexts, we use techniques from [1, 8, 11] to estimate the weight matrix $W_*$, and define "optimistic estimates" of $W_*$. We also use the technique of combining the objective and the constraints using a certain tradeoff parameter and that was introduced in [4]. Further new difficulties arise, such as in estimating the optimum value from the first few rounds, a task that follows from standard techniques in each of the special cases but is very challenging here. We develop a new way of exploration that uses the linear structure, so that one can evaluate all possible choices that could have led to an optimum solution on the historic sample. This technique might be of independent interest in estimating optimum values. One can see that the problem is indeed more than the sum of its parts, from the fact that we get the optimal bound for linCBwK only when $B \geq \tilde{\Omega}(m^{1/2}T^{3/4})$, unlike either special case for which the optimal bound holds for all $B$ (but is meaningful only for $B = \tilde{\Omega}(m\sqrt{T})$).

The approach in [3] (for BwK) extends to the case of "static" contexts,[4] where each arm has a context that doesn't change over time. The OSPP of [4] is *not* a special case of linCBwK with static contexts; this is one indication of the additional difficulty of dynamic over static contexts.

**Other related work.** Recently, [17] showed an $O(\sqrt{T})$ regret in the linear contextual setting with a single budget constraint, when costs depend only on contexts and not arms.

Due to space constraints, we have moved many proofs from the main part of the paper to the supplement.

## 2 Preliminaries

### 2.1 Confidence Ellipsoid

Consider a stochastic process which in each round $t$, generates a pair of observations $(r_t, \boldsymbol{y}_t)$, such that $r_t$ is an unknown linear function of $\boldsymbol{y}_t$ plus some 0-mean bounded noise, i.e., $r_t = \boldsymbol{\mu}_*^\top \boldsymbol{y}_t + \eta_t$, where $\boldsymbol{y}_t, \boldsymbol{\mu}_* \in \mathbb{R}^m$, $|\eta_t| \leq 2R$, and $\mathbb{E}[\eta_t | \boldsymbol{y}_1, r_1, \ldots, \boldsymbol{y}_{t-1}, r_{t-1}, \boldsymbol{y}_t] = 0$.

At any time $t$, a high confidence estimate of the unknown vector $\boldsymbol{\mu}_*$ can be obtained by building a "confidence ellipsoid" around the $\ell_2$-regularized least-squares estimate $\hat{\boldsymbol{\mu}}_t$ constructed from the observations made so far. This technique is common in prior work on linear contextual bandits (e.g., in [8, 11, 1]). For any regularization parameter $\lambda > 0$, let

$$M_t := \lambda I + \sum_{i=1}^{t-1} \boldsymbol{y}_i \boldsymbol{y}_i^\top, \text{ and } \hat{\boldsymbol{\mu}}_t := M_t^{-1} \sum_{i=1}^{t-1} \boldsymbol{y}_i r_i.$$

The following result from [1] shows that $\boldsymbol{\mu}_*$ lies with high probability in an ellipsoid with center $\hat{\boldsymbol{\mu}}_t$. For any positive semi-definite (PSD) matrix $M$, define the $M$-norm as $\|\boldsymbol{\mu}\|_M := \sqrt{\boldsymbol{\mu}^\top M \boldsymbol{\mu}}$. The confidence ellipsoid at time $t$ is defined as

$$C_t := \left\{ \boldsymbol{\mu} \in \mathbb{R}^m : \|\boldsymbol{\mu} - \hat{\boldsymbol{\mu}}_t\|_{M_t} \leq R\sqrt{m \log\left((1+tm/\lambda)/\delta\right)} + \sqrt{\lambda m} \right\}.$$

**Lemma 2** (Theorem 2 of [1]). *If $\forall\, t$, $\|\boldsymbol{\mu}_*\|_2 \leq \sqrt{m}$ and $\|\boldsymbol{y}_t\|_2 \leq \sqrt{m}$, then with prob. $1 - \delta$, $\boldsymbol{\mu}_* \in C_t$.*

Another useful observation about this construction is stated below. It first appeared as Lemma 11 of [8], and was also proved as Lemma 3 in [11].

**Lemma 3** (Lemma 11 of [8]). $\sum_{t=1}^{T} \|\boldsymbol{y}_t\|_{M_t^{-1}} \leq \sqrt{mT \log(T)}$.

As a corollary of the above two lemmas, we obtain a bound on the total error in the estimate provided by "any point" from the confidence ellipsoid. (Proof is in Appendix D in the supplement.)

**Corollary 1.** *For $t = 1, \ldots, T$, let $\tilde{\boldsymbol{\mu}}_t \in C_t$ be a point in the confidence ellipsoid, with $\lambda = 1$ and $2R = 1$. Then, with probability $1 - \delta$,*

$$\textstyle\sum_{t=1}^{T} |\tilde{\boldsymbol{\mu}}_t^\top \boldsymbol{y}_t - \boldsymbol{\mu}_*^\top \boldsymbol{y}_t| \leq 2m\sqrt{T \log\left((1+Tm)/\delta\right) \log(T)}.$$

## 2.2 Online Learning

Consider a $T$ round game played between an online learner and an adversary, where in round $t$, the learner chooses a $\boldsymbol{\theta}_t \in \Omega := \{\boldsymbol{\theta} : \|\boldsymbol{\theta}\|_1 \leq 1, \boldsymbol{\theta} \geq 0\}$, and then observes a linear function $g_t : \Omega \to [-1, 1]$ picked by the adversary. The learner's choice $\boldsymbol{\theta}_t$ may only depend on learner's and adversary's choices in previous rounds. The goal of the learner is to minimize *regret* defined as the difference between the learner's objective value and the value of the best single choice in hindsight:

$$\mathcal{R}(T) := \max_{\boldsymbol{\theta} \in \Omega} \textstyle\sum_{t=1}^{T} g_t(\boldsymbol{\theta}) - \sum_{t=1}^{T} g_t(\boldsymbol{\theta}_t).$$

The *multiplicative weight update* (MWU) algorithm (generalization by [7]) is a fast and efficient online learning algorithm for this problem. Let $g_{t,j} := g_t(\mathbf{1}_j)$. Then, given a parameter $\epsilon > 0$, in round $t + 1$, the choice of this algorithm takes the following form,

$$\boldsymbol{\theta}_{t+1,j} = \frac{w_{t,j}}{1 + \sum_j w_{t,j}}, \text{ where } w_{t,j} = \begin{cases} w_{t-1,j}(1+\epsilon)^{g_{t,j}} & \text{if } g_{t,j} > 0, \\ w_{t-1,j}(1-\epsilon)^{-g_{t,j}} & \text{if } g_{t,j} \leq 0. \end{cases} \tag{5}$$

with initialization $w_{0,j} = 1$, for all $j = 1, \ldots, K$.

**Lemma 4.** *[7] For any $0 < \epsilon \leq \frac{1}{2}$, the MWU algorithm provides the following regret bound for the online learning problem described above:*

$$\mathcal{R}(T) \leq \epsilon T + \tfrac{\log(d+1)}{\epsilon}.$$

*In particular, for $\epsilon = \sqrt{\frac{\log(d+1)}{T}}$, we have $\mathcal{R}(T) \leq \sqrt{\log(d+1)T}$*

For the rest of the paper, we refer to the MWU algorithm with $\epsilon = \sqrt{\frac{\log(d+1)}{T}}$ as the online learning (OL) algorithm, and the update in (5) as the OL update at time $t + 1$.

# 3 Algorithm

## 3.1 Optimistic estimates of unknown parameters

Let $a_t$ denote the arm played by the algorithm at time $t$. In the beginning of every round, we use the outcomes and contexts from previous rounds to construct a confidence ellipsoid for $\boldsymbol{\mu}_*$ and every column of $W_*$. The construction of confidence ellipsoid for $\boldsymbol{\mu}_*$ follows directly from the techniques in Section 2.1 with $y_t = \mathbf{x}_t(a_t)$ and $r_t$ being reward at time $t$. To construct a confidence ellipsoid for a column $j$ of $W_*$, we use the techniques in Section 2.1 while substituting $\boldsymbol{y}_t = \mathbf{x}_t(a_t)$ and $r_t = \mathbf{v}_t(a_t)_j$ for every $j$.

As in Section 2.1, let $M_t := I + \sum_{i=1}^{t-1} \mathbf{x}_i(a_i)\mathbf{x}_i(a_i)^\top$, and construct the regularized least squares estimate for $\boldsymbol{\mu}_*, W_*$, respectively, as

$$\hat{\boldsymbol{\mu}}_t \quad := \quad M_t^{-1} \textstyle\sum_{i=1}^{t-1} \mathbf{x}_i(a_i) r_i(a_i)^\top \tag{6}$$

$$\hat{W}_t \quad := \quad M_t^{-1} \textstyle\sum_{i=1}^{t-1} \mathbf{x}_i(a_i) \mathbf{v}_i(a_i)^\top. \tag{7}$$

Define confidence ellipsoid for parameter $\boldsymbol{\mu}_*$ as

$$C_{t,0} := \left\{ \boldsymbol{\mu} \in \mathbb{R}^m : \|\boldsymbol{\mu} - \hat{\boldsymbol{\mu}}\|_{M_t} \leq \sqrt{m \log\left((d+tmd)/\delta\right)} + \sqrt{m} \right\},$$

and for every arm $a$, the optimistic estimate of $\boldsymbol{\mu}_*$ as:

$$\tilde{\boldsymbol{\mu}}_t(a) := \arg\max_{\boldsymbol{\mu} \in C_{t,0}} \mathbf{x}_t(a)^\top \boldsymbol{\mu}. \tag{8}$$

Let $\mathbf{w}_j$ denote the $j^{th}$ column of a matrix $W$. We define a confidence ellipsoid for each column $j$, as

$$C_{t,j} := \left\{ \mathbf{w} \in \mathbb{R}^m : \|\mathbf{w} - \hat{\mathbf{w}}_{tj}\|_{M_t} \leq \sqrt{m \log\left((d+tmd)/\delta\right)} + \sqrt{m} \right\},$$

and denote by $\mathcal{G}_t$, the Cartesian product of all these ellipsoids: $\mathcal{G}_t := \{W \in \mathbb{R}^{m \times d} : \mathbf{w}_j \in C_{t,j}\}$. Note that Lemma 2 implies that $W_* \in \mathcal{G}_t$ with probability $1 - \delta$. Now, given a vector $\boldsymbol{\theta}_t \in \mathbb{R}^d$, we define the *optimistic estimate* of the weight matrix at time $t$ w.r.t. $\boldsymbol{\theta}_t$, for every arm $a \in [K]$, as :

$$\tilde{W}_t(a) := \arg\min_{W \in \mathcal{G}_t} \mathbf{x}_t(a)^\top W \boldsymbol{\theta}_t. \tag{9}$$

Intuitively, for the reward, we want an upper confidence bound and for the consumption we want a lower confidence bound as an optimistic estimate. This intuition aligns with the above definitions, where the maximizer was used in case of reward and a minimizer was used for consumption. The utility and precise meaning of $\boldsymbol{\theta}_t$ will become clearer when we describe the algorithm and present the regret analysis.

Using the definition of $\tilde{\boldsymbol{\mu}}_t, \tilde{W}_t$, along with the results in Lemma 2 and Corollary 1 about confidence ellipsoids, the following can be derived.

**Corollary 2.** *With probability $1 - \delta$, for any sequence of $\boldsymbol{\theta}_1, \boldsymbol{\theta}_2, \ldots, \boldsymbol{\theta}_T$,*

1. $\mathbf{x}_t(a)^\top \tilde{\boldsymbol{\mu}}_t(a) \geq \mathbf{x}_t(a)^\top \boldsymbol{\mu}_*$, *for all arms $a \in [K]$, for all time $t$.*

2. $\mathbf{x}_t(a)^\top \tilde{W}_t(a) \boldsymbol{\theta}_t \leq \mathbf{x}_t(a)^\top W_* \boldsymbol{\theta}_t$, *for all arms $a \in [K]$, for all time $t$.*

3. $|\sum_{t=1}^{T} (\tilde{\boldsymbol{\mu}}_t(a_t) - \boldsymbol{\mu}_*)^\top \mathbf{x}_t(a_t)| \leq \left(2m\sqrt{T \log\left((1+tm)/\delta\right) \log(T)}\right).$

4. $\|\sum_{t=1}^{T} (\tilde{W}_t(a_t) - W_*)^\top \mathbf{x}_t(a_t)\| \leq \|\mathbf{1}_d\| \left(2m\sqrt{T \log\left((d+tmd)/\delta\right) \log(T)}\right).$

Essentially, the first two claims ensure that we have optimistic estimates, and the last two claims ensure that the estimates quickly converge to the true parameters.

## 3.2 The core algorithm

In this section, we present an algorithm and its analysis, under the assumption that a parameter $Z$ satisfying certain properties is given. Later, we show how to use the first $T_0$ rounds to compute such a $Z$, and also bound the additional regret due to these $T_0$ rounds. We define $Z$ now.

**Assumption 1.** *Let $Z$ be such that for some universal constants $c, c'$, $\frac{OPT}{B} \leq Z \leq c\frac{OPT}{B} + c'$.*

The algorithm constructs estimates $\hat{\boldsymbol{\mu}}_t$ and $\hat{W}_t$ as in Section 3.1. It also runs the OL algorithm for an instance of the online learning problem. The vector played by the OL algorithm in time step $t$ is $\boldsymbol{\theta}_t$. After observing the context, the optimistic estimates for each arm are then constructed using $\boldsymbol{\theta}_t$, as defined in (8) and (9). Intuitively, $\boldsymbol{\theta}_t$ is used here as a multiplier to combine different columns of the weight matrix, to get an optimistic weight vector for every arm. An *adjusted estimated reward* for arm $a$ is then defined by using $Z$ to linearly combine the optimistic estimate of the reward with the optimistic estimate of the consumption, as $(\mathbf{x}_t(a)^\top \tilde{\boldsymbol{\mu}}_t(a)) - Z(\mathbf{x}_t(a)^\top \tilde{W}_t(a) \boldsymbol{\theta}_t)$. The algorithm chooses the arm which appears to be the best according to the adjusted estimated reward. After observing the resulting reward and consumption vectors, the estimates are updated. The online learning algorithm is advanced by one step, by defining the profit vector to be $\mathbf{v}_t(a_t) - \frac{B}{T}\mathbf{1}$. The algorithm ends either after $T$ time steps or as soon as the total consumption exceeds the budget along some dimension.

**Theorem 2.** *Given a $Z$ as per Assumption 1, Algorithm 1 achieves the following, with prob. $1 - \delta$:*

$$regret(T) \leq O\left(\left(\frac{OPT}{B} + 1\right)m\sqrt{T \log(dT/\delta) \log(T)}\right).$$

*(Proof Sketch)* We provide a sketch of the proof here, with a full proof given in Appendix E in the supplement. Let $\tau$ be the stopping time of the algorithm. The proof is in 3 steps:

**Step 1:** Since $\mathbb{E}[\mathbf{v}_t(a_t)|X_t, a_t, H_{t-1}] = W_*^\top \mathbf{x}_t(a_t)$, we apply Azuma-Hoeffding inequality to get that with high probability $\left\|\sum_{t=1}^{\tau} \mathbf{v}_t(a_t) - W_*^\top \mathbf{x}_t(a_t)\right\|_\infty$ is small. Therefore, we can work with $\sum_{t=1}^{\tau} W_*^\top \mathbf{x}_t(a_t)$ instead of $\sum_{t=1}^{\tau} \mathbf{v}_t(a_t)$. A similar application of Azuma-Hoeffding inequality is used to bound the gap $|\sum_{t=1}^{\tau} r_t(a_t) - \boldsymbol{\mu}_*^\top \mathbf{x}_t(a_t)|$, so that a lower bound on $\sum_{t=1}^{\tau} \boldsymbol{\mu}_*^\top \mathbf{x}_t(a_t)$ is sufficient to lower bound the total reward $\sum_{t=1}^{\tau} r_t(a_t)$.

---

**Algorithm 1** Algorithm for linCBwK, with given $Z$

---

Initialize $\boldsymbol{\theta}_1$ as per the online learning (OL) algorithm.
Initialize $Z$ that satisfies Assumption 1.
**for all** $t = 1, ..., T$ **do**
    Observe $X_t$.
    For every $a \in [K]$, compute $\tilde{\boldsymbol{\mu}}_t(a)$ and $\tilde{W}_t(a)$ as per (8) and (9) respectively.
    Play the arm $a_t := \arg\max_{a \in [K]} \mathbf{x}_t(a)^\top (\tilde{\boldsymbol{\mu}}_t(a) - Z\tilde{W}_t(a)\boldsymbol{\theta}_t)$.
    Observe $r_t(a_t)$ and $\mathbf{v}_t(a_t)$.
    If for some $j = 1..d$, $\sum_{t' \le t} \mathbf{v}_{t'}(a_{t'}) \cdot \mathbf{e}_j \ge B$ then EXIT.
    Use $\mathbf{x}_t(a_t), r_t(a_t)$ and $\mathbf{v}_t(a_t)$ to obtain $\hat{\boldsymbol{\mu}}_{t+1}, \hat{W}_{t+1}$ and $\mathcal{G}_{t+1}$.
    Choose $\boldsymbol{\theta}_{t+1}$ using the OL update (refer to (5)) with $g_t(\boldsymbol{\theta}_t) := \boldsymbol{\theta}_t \cdot \left(\mathbf{v}_t(a_t) - \frac{B}{T}\mathbf{1}\right)$.
**end for**

---

**Step 2:** Using Corollary 2, with high probability, we can bound $\left\|\sum_{t=1}^T (W_* - \tilde{W}_t(a_t))^\top \mathbf{x}_t(a_t)\right\|_\infty$.
It is therefore sufficient to work with the sum of vectors $\tilde{W}_t(a_t)^\top \mathbf{x}_t(a_t)$ instead of $W_*^\top \mathbf{x}_t(a_t)$, and similarly with $\tilde{\boldsymbol{\mu}}_t(a_t)^\top \mathbf{x}_t(a_t)$ instead of $\mu_*^\top \mathbf{x}_t(a_t)$.

**Step 3:** The proof is completed by showing the desired bound on OPT $- \sum_{t=1}^\tau \tilde{\boldsymbol{\mu}}_t(a_t)^\top \mathbf{x}_t(a_t)$. This part is similar to the online stochastic packing problem; if the actual reward and consumption vectors were $\tilde{\boldsymbol{\mu}}_t(a_t)^\top \mathbf{x}_t(a_t)$ and $\tilde{W}_t(a_t)^\top \mathbf{x}_t(a_t)$, then it would be exactly that problem. We adapt techniques from [4]: use the OL algorithm and the $Z$ parameter to combine constraints into the objective. If a dimension is being consumed too fast, then the multiplier for that dimension should increase, making the algorithm to pick arms that are not likely to consume too much along this dimension. Regret is then bounded by a combination of the online learning regret and the error in the optimistic estimates.

### 3.3 Algorithm with $Z$ computation

In this section, we present a modification of Algorithm 1 which computes the required parameter $Z$ that satisfies Assumption 1, and therefore does not need to be provided with a $Z$ as input. The algorithm computes $Z$ using observations from the first $T_0$ rounds. Once $Z$ is computed, Algorithm 1 can be run for the remaining time steps. However, it needs to be modified slightly to take into account the budget consumed during the first $T_0$ rounds. We handle this by using a smaller budget $B' = B - T_0$ in the computations for the remaining rounds. The modified algorithm is given below.

---

**Algorithm 2** Algorithm for linCBwK, with $Z$ computation

---

**Inputs:** $B, T_0, B' = B - T_0$
Using observations from first $T_0$ rounds, compute a $Z$ that satisfies Assumption 1.
Run Algorithm 1 for $T - T_0$ rounds and budget $B'$.

---

Next, we provide the details of how to compute $Z$ from observations in the first $T_0$ rounds, and how to choose $T_0$. We provide a method that takes advantage of the linear structure of the problem, and explores in the $m$-dimensional space of contexts and weight vectors to obtain bounds independent of $K$. In every round $t = 1, \ldots, T_0$, after observing $X_t$, let $p_t \in \Delta^{[K]}$ be

$$p_t := \arg\max_{p \in \Delta^{[K]}} \|X_t p\|_{M_t^{-1}}, \tag{10}$$

$$\text{where} \quad M_t := I + \sum_{i=1}^{t-1} (X_i p_i)(X_i p_i)^\top. \tag{11}$$

Select arm $a_t = a$ with probability $p_t(a)$. In fact, since $M_t$ is a PSD matrix, due to convexity of the function $\|X_t p\|_{M_t^{-1}}^2$, it is the same as playing $a_t = \arg\max_{a \in [K]} \|\mathbf{x}_t(a)\|_{M_t^{-1}}$. Construct estimates $\hat{\boldsymbol{\mu}}, \hat{W}_t$ of $\boldsymbol{\mu}_*, W_*$ at time $t$ as

$$\hat{\boldsymbol{\mu}}_t := M_t^{-1} \sum_{i=1}^{t-1} (X_i p_i) r_i(a_i), \quad \hat{W}_t := M_t^{-1} \sum_{i=1}^{t-1} (X_i p_i) \mathbf{v}_i(a_i)^\top.$$

And, for some value of $\gamma$ defined later, obtain an estimate $\hat{\text{OPT}}^{\gamma}$ of OPT as:

$$\hat{\text{OPT}}^{\gamma} := \begin{array}{c} \max_{\pi} \\ \text{such that} \end{array} \quad \begin{array}{c} \frac{T}{T_0} \sum_{i=1}^{T_0} \hat{\boldsymbol{\mu}}_i^{\top} X_i \pi(X_i) \\ \frac{T}{T_0} \sum_{i=1}^{T_0} \hat{W}_i^{\top} X_i \pi(X_i) \leq B + \gamma. \end{array} \tag{12}$$

For an intuition about the choice of arm in (10), observe from the discussion in Section 2.1 that every column $\mathbf{w}_{*j}$ of $W_*$ is guaranteed to lie inside the confidence ellipsoid centered at column $\hat{\mathbf{w}}_{tj}$ of $\hat{W}_t$, namely the ellipsoid, $\|\mathbf{w} - \hat{\mathbf{w}}_{tj}\|_{M_t}^2 \leq 4m \log(Tm/\delta)$. Note that this ellipsoid has principle axes as eigenvectors of $M_t$, and the length of the semi-principle axes is given by the *inverse* eigenvalues of $M_t$. Therefore, by maximizing $\|X_t p\|_{M_t^{-1}}$ we are choosing the context closest to the direction of the longest principal axis of the confidence ellipsoid, i.e. in the direction of the maximum uncertainty. Intuitively, this corresponds to pure exploration: by making an observation in the direction where uncertainty is large we can reduce the uncertainty in our estimate most effectively.

A more algebraic explanation is as follows. In order to get a good estimate of OPT by $\hat{\text{OPT}}^{\gamma}$, we want the estimates $\hat{W}_t$ and $W_*$ (and, $\hat{\boldsymbol{\mu}}$ and $\boldsymbol{\mu}_*$) to be close enough so that $\|\sum_{t=1}^{T_0}(\hat{W}_t - \hat{W}_*)^{\top} X_t \pi(X_t)\|_{\infty}$ (and, $|\sum_{t=1}^{T_0}(\hat{\boldsymbol{\mu}}_t - \boldsymbol{\mu}_*)^{\top} X_t \pi(X_t)|$) is small for all policies $\pi$, and in particular for sample optimal policies. Now, using Cauchy-Schwartz these are bounded by

$$\sum_{t=1}^{T_0} \|\hat{\boldsymbol{\mu}}_t - \boldsymbol{\mu}_*\|_{M_t} \|X_t \pi(X_t))\|_{M_t^{-1}}, \text{ and}$$

$$\sum_{t=1}^{T_0} \|\hat{W}_t - W_*\|_{M_t} \|X_t \pi(X_t))\|_{M_t^{-1}},$$

where we define $\|W\|_M$, the $M$-norm of matrix $W$ to be the max of column-wise $M$-norms. Using Lemma 2, the term $\|\hat{\boldsymbol{\mu}}_t - \boldsymbol{\mu}_*\|_{M_t}$ is bounded by $2\sqrt{m \log(T_0 m/\delta)}$, and $\|\hat{W}_t - W_*\|_{M_t}$ is bounded by $2\sqrt{m \log(T_0 m d/\delta)}$, with probability $1 - \delta$. Lemma 3 bounds the second term $\sum_{t=1}^{T_0} \|X_t \pi(X_t)\|_{M_t^{-1}}$ but only when $\pi$ is the played policy. This is where we use that the played policy $p_t$ was chosen to maximize $\|X_t p_t\|_{M_t^{-1}}$, so that $\sum_{t=1}^{T_0} \|X_t \pi(X_t)\|_{M_t^{-1}} \leq \sum_{t=1}^{T_0} \|X_t p_t\|_{M_t^{-1}}$ and the bound $\sum_{t=1}^{T_0} \|X_t p_t\|_{M_t^{-1}} \leq \sqrt{m T_0 \log(T_0)}$ given by Lemma 3 actually bounds $\sum_{t=1}^{T_0} \|X_t \pi(X_t)\|_{M_t^{-1}}$ for all $\pi$. Combining, we get a bound of $2m\sqrt{T_0 \log(T_0) \log(T_0 d/\delta)}$ on deviations $\|\sum_{t=1}^{T_0}(\hat{W}_t - \hat{W}_*)^{\top} X_t \pi(X_t)\|_{\infty}$ and $|\sum_{t=1}^{T_0}(\hat{\boldsymbol{\mu}}_t - \boldsymbol{\mu}_*)^{\top} X_t \pi(X_t)|$ for all $\pi$.

We prove the following lemma.

**Lemma 5.** *For* $\gamma = \left(\frac{T}{T_0}\right) 2m\sqrt{T_0 \log(T_0) \log(T_0 d/\delta)}$, *with probability* $1 - O(\delta)$,

$$OPT - 2\gamma \leq \hat{OPT}^{2\gamma} \leq OPT + 9\gamma(\frac{OPT}{B} + 1).$$

**Corollary 3.** *Set* $Z = \frac{(\hat{OPT}^{2\gamma} + 2\gamma)}{B} + 1$, *with the above value of* $\gamma$. *Then, with probability* $1 - O(\delta)$,

$$\frac{OPT}{B} + 1 \leq Z \leq (1 + \frac{11\gamma}{B})(\frac{OPT}{B} + 1).$$

Corollary 3 implies that as long as $B \geq \gamma$, i.e., $B \geq \tilde{\Omega}(\frac{mT}{\sqrt{T_0}})$, $Z$ is a constant factor approximation of $\frac{OPT}{B} + 1 \geq Z^*$, therefore Theorem 2 should provide an $\tilde{O}\left((\frac{OPT}{B} + 1)m\sqrt{T}\right)$ regret bound. However, this bound does not account for the budget consumed in the first $T_0$ rounds. Considering that (at most) $T_0$ amount can be consumed from the budget in the first $T_0$ rounds, we have an additional regret of $\frac{OPT}{B} T_0$. Further, since we have $B' = B - T_0$ budget for remaining $T - T_0$ rounds, we need a $Z$ that satisfies the required assumption for $B'$ instead of $B$ (i.e., we need $\frac{OPT}{B'} \leq Z \leq O(1)\left(\frac{OPT}{B'} + 1\right)$). If $B \geq 2T_0$, then, $B' \geq B/2$, and using 2 times the $Z$ computed in Corollary 3 would satisfy the required assumption.

Together, these observations give Theorem 3.

**Theorem 3.** *Using Algorithm 2 with* $T_0$ *such that* $B > \max\{2T_0, mT/\sqrt{T_0}\}$, *and twice the* $Z$ *given by Corollary 3, we get a high probability regret bound of*

$$\tilde{O}\left(\left(\frac{OPT}{B} + 1\right)\left(T_0 + m\sqrt{T}\right)\right).$$

*In particular, for* $B > m^{1/2}T^{3/4}$ *and* $m \leq \sqrt{T}$, *we can use* $T_0 = m\sqrt{T}$ *to get a regret bound of*

$$\tilde{O}\left(\left(\frac{OPT}{B} + 1\right) m\sqrt{T}\right).$$

## Footnotes

[3]Similar to the regret bounds for linear contextual bandits [8, 1, 11].

[4]It was incorrectly claimed in [3] that the approach can be extended to dynamic contexts without much modifications.

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
