[Supplementary Material]



## A Concentration Inequalities

**Lemma 6** (Azuma-Hoeffding inequality). *If a super-martingale $(Y_t; t \geq 0)$, corresponding to filtration $\mathcal{F}_t$, satisfies $|Y_t - Y_{t-1}| \leq c_t$ for some constant $c_t$, for all $t = 1, \ldots, T$, then for any $a \geq 0$,*

$$\Pr(Y_T - Y_0 \geq a) \leq e^{-\frac{a^2}{2\sum_{t=1}^{T} c_t^2}}.$$

## B Benchmark

*Proof of Lemma 1.* For an instantiation $\omega = (X_t, V_t)_{t=1}^{T}$ of the sequence of inputs, let vector $\mathbf{p}_t^*(\omega) \in \Delta^{K+1}$ denote the distribution over actions (plus no-op) taken by the *optimal adaptive policy* at time $t$. Then,

$$\overline{\text{OPT}} = \mathbb{E}_{\omega \sim \mathcal{D}^T}[\sum_{t=1}^{T} \mathbf{r}_t^\top \mathbf{p}_t^*(\omega)] \tag{13}$$

Also, since this is a feasible policy,

$$\mathbb{E}_{\omega \sim \mathcal{D}^T}[\sum_{t=1}^{T} V_t^\top \mathbf{p}_t^*(\omega)] \leq B\mathbf{1} \tag{14}$$

Construct a *static* context dependent policy $\pi^*$ as follows: for any $X \in [0,1]^{m \times K}$, define

$$\pi^*(X) := \frac{1}{T} \sum_{t=1}^{T} \mathbb{E}_\omega[\mathbf{p}_t^*(\omega)|X_t = X].$$

Intuitively, $\pi^*(X)_a$ denotes (in hindsight) the probability that the optimal adaptive policy takes an action $a$ when presented with a context $X$, averaged over all time steps. Now, by definition of $\mathbf{r}(\pi), \mathbf{v}(\pi)$, from above definition of $\pi^*$, and (13), (14),

$$T\mathbf{r}(\pi^*) = T\mathbb{E}_{X \sim \mathcal{D}}[\boldsymbol{\mu}_*^\top X \pi^*(X)] = \mathbb{E}_\omega[\sum_{t=1}^{T} V_t \mathbf{p}_t^*(\omega)] = \overline{\text{OPT}},$$
$$T\mathbf{v}(\pi^*) = T\mathbb{E}_{X \sim \mathcal{D}}[W_*^\top X \pi^*(X)] = \mathbb{E}_\omega[\sum_{t=1}^{T} V_t \mathbf{p}_t^*(\omega)] \leq B\mathbf{1},$$

$\square$

## C Hardness of linear AMO

In this section we show that finding the best linear policy is NP-Hard. The input to the problem is, for each $t \in [T]$, and each arm $a \in [K]$, a context $\mathbf{x}_t(a) \in [0,1]^m$, and a reward $r_t(a) \in [-1,1]$. The output is a vector $\boldsymbol{\theta} \in \Re^m$ that maximizes $\sum_t r_t(a_t)$ where

$$a_t = \arg\max_{a \in [K]} \{\mathbf{x}_t(a)^\top \theta\}.$$

We give a reduction from the problem of learning halfspaces with noise [16]. The input to this problem is for some integer $n$, for each $i \in [n]$, a vector $z_i \in [0,1]^m$, and $y_i \in \{-1, +1\}$. The output is a vector $\boldsymbol{\theta} \in \Re^m$ that maximizes

$$\sum_{i=1}^{n} sign(\mathbf{z}_i^\top \boldsymbol{\theta}) y_i.$$

Given an instance of the problem of learning halfspaces with noise, construct an instance of the linear AMO as follows. The time horizon $T = n$, and the number of arms $K = 2$. For each $t \in [T]$, the context of the first arm, $\mathbf{x}_t(1) = z_t$, and its reward $r_t(1) = y_t$. The context of the second arm, $\mathbf{x}_t(2) = \mathbf{0}$, the all zeroes vector, and the reward $r_t(2)$ is also 0.

The total reward of a linear policy w.r.t a vector $\boldsymbol{\theta}$ for this instance is

$$|\{i : sign(\mathbf{z}_i^\top \boldsymbol{\theta}) = 1, y_i = 1\}| - |\{i : sign(\mathbf{z}_i^\top \boldsymbol{\theta}) = 1, y_i = -1\}|.$$

It is easy to see that this is an affine transformation of the objective for the problem of learning halfspaces with noise.

## D Confidence ellipsoids

**Proof of Corollary 1.** The following holds with probability $1 - \delta$.

$$
\sum_{t=1}^{T} |\tilde{\boldsymbol{\mu}}_t^\top \mathbf{x}_t - \boldsymbol{\mu}_*^\top \mathbf{x}_t| \leq \sum_{t=1}^{T} \|\tilde{\boldsymbol{\mu}}_t - \boldsymbol{\mu}_*\|_{M_t} \|\mathbf{x}_t\|_{M_t^{-1}}
$$

$$
\leq \left( \sqrt{m \ln\left( \frac{1 + tm}{\delta} \right)} + \sqrt{m} \right) \sqrt{mT \ln(T)}.
$$

The inequality in the first line is a matrix-norm version of Cauchy-Schwartz (Lemma 7). The inequality in the second line is due to Lemmas 2 and 3. The lemma follows from multiplying out the two factors in the second line.

$\square$

**Lemma 7.** *For any positive definite matrix $M \in \mathbb{R}^{n \times n}$ and any two vectors $\mathbf{a}, \mathbf{b} \in \mathbb{R}^n$, $|\mathbf{a}^\top \mathbf{b}| \leq \|\mathbf{a}\|_M \|\mathbf{b}\|_{M^{-1}}$.*

*Proof.* Since $M$ is positive definite, there exists a matrix $M_{1/2}$ such that $M = M_{1/2} M_{1/2}^\top$. Further, $M^{-1} = M_{-1/2}^\top M_{-1/2}$ where $M_{-1/2} = M_{1/2}^{-1}$.

$$
\|\mathbf{a}^\top M_{1/2}\|^2 = \mathbf{a}^\top M_{1/2} M_{1/2}^\top \mathbf{a} = \mathbf{a}^\top M \mathbf{a} = \|\mathbf{a}\|_M^2.
$$

Similarly, $\|M_{-1/2} \mathbf{b}\|^2 = \|\mathbf{b}\|_{M^{-1}}^2$. Now applying Cauchy-Schwartz, we get that

$$
|\mathbf{a}^\top \mathbf{b}| = |\mathbf{a}^\top M_{1/2} M_{-1/2} \mathbf{b}| \leq \|\mathbf{a}^\top M_{1/2}\| \|M_{-1/2} \mathbf{b}\| = \|\mathbf{a}\|_M \|\mathbf{b}\|_{M^{-1}}.
$$

$\square$

**Proof of Corollary 2.** Here, the first claim follows simply from definition of $\tilde{W}_t(a)$ and the observation that with probability $1 - \delta$, $W^* \in \mathcal{G}_t$. To obtain the second claim, apply Corollary 1 with $\boldsymbol{\mu}_* = \mathbf{w}_{*j}, \boldsymbol{y}_t = \mathbf{x}_t(a_t), \tilde{\boldsymbol{\mu}}_t = [\tilde{W}_t(a_t)]_j$ (the $j^{th}$ column of $\tilde{W}_t(a_t)$), to bound $|\sum_t ([\tilde{W}_t(a_t)]_j - \mathbf{w}_{*j})^\top \mathbf{x}_t(a_t)| \leq \sum_t |([\tilde{W}_t(a_t)]_j - \mathbf{w}_{*j})^\top \mathbf{x}_t(a_t)|$ for every $j$, and then take the norm. $\square$

## E Appendix for Section 3.2

**Proof of Theorem 2:** We will use $\mathcal{R}'$ to denote the main term in the regret bound.

$$
\mathcal{R}'(T) := O\left( m \sqrt{\ln(mdT/\delta) \ln(T) T} \right)
$$

Let $\tau$ be the stopping time of the algorithm. Let $H_{t-1}$ be the history of plays and observations before time $t$, i.e. $H_{t-1} := \{\boldsymbol{\theta}_\tau, X_\tau, a_\tau, r_\tau(a_\tau), \mathbf{v}_\tau(a_\tau), \tau = 1, \ldots, t-1\}$. Note that $H_{t-1}$ determines $\boldsymbol{\theta}_t, \hat{\boldsymbol{\mu}}_t, \hat{W}_t, \mathcal{G}_t$, but it does not determine $X_t, a_t, \tilde{W}_t$ (since $a_t$ and $\tilde{W}_t(a)$ depend on the context $X_t$ at time $t$). The proof is in 3 steps:

**Step 1:** Since $\mathbb{E}[\mathbf{v}_t(a_t)|X_t, a_t, H_{t-1}] = W_*^\top \mathbf{x}_t(a_t)$, we apply Azuma-Hoeffding inequality to get that with probability $1 - \delta$,

$$
\left\| \sum_{t=1}^{\tau} \mathbf{v}_t(a_t) - W_*^\top \mathbf{x}_t(a_t) \right\|_\infty \leq \mathcal{R}'(T). \tag{15}
$$

Similarly, we obtain

$$
\left| \sum_{t=1}^{\tau} r_t(a_t) - \boldsymbol{\mu}_*^\top \mathbf{x}_t(a_t) \right| \leq \mathcal{R}'(T). \tag{16}
$$

**Step 2:** From Corollary 2, with probability $1 - \delta$,

$$\left\| \sum_{t=1}^{T} (W_* - \tilde{W}_t(a_t))^\top \mathbf{x}_t(a_t) \right\|_\infty \leq \mathcal{R}'(T). \tag{17}$$

$$| \sum_{t=1}^{T} (\tilde{\boldsymbol{\mu}}_t(a_t) - \boldsymbol{\mu}_*)^\top \mathbf{x}_t(a_t) | \leq \mathcal{R}'(T). \tag{18}$$

It is therefore sufficient to bound the sum of the vectors $\tilde{W}_t(a_t)^\top \mathbf{x}_t(a_t)$, and similarly for $\tilde{\boldsymbol{\mu}}_t(a_t)^\top \mathbf{x}_t(a_t)$. We use the shorthand notation of $\tilde{r}_t := \tilde{\boldsymbol{\mu}}_t(a_t)^\top \mathbf{x}_t(a_t)$, $\tilde{r}_{\text{sum}} := \sum_{t=1}^{\tau} \tilde{r}_t$, $\tilde{\mathbf{v}}_t := \tilde{W}_t(a_t)^\top \mathbf{x}_t(a_t)$ and $\tilde{\mathbf{v}}_{\text{sum}} := \sum_{t=1}^{\tau} \tilde{\mathbf{v}}_t$ for the rest of this proof.

**Step 3:** The proof is completed by showing that

$$\mathbb{E}[\tilde{r}_{\text{sum}}] \geq \text{OPT} - Z\mathcal{R}'(T).$$

**Lemma 8.**

$$\sum_{t=1}^{\tau} \mathbb{E}[\tilde{r}_t | H_{t-1}] \geq \frac{\tau}{T} OPT + Z \sum_{t=1}^{\tau} \boldsymbol{\theta}_t \cdot \mathbb{E}[\tilde{\mathbf{v}}_t - \mathbf{1}\frac{B}{T} | H_{t-1}]$$

*Proof.* Let $a_t^*$ be defined as the (randomized) action given by optimal static policy $\pi^*$ for context $X_t$. Define $r_t^* := \boldsymbol{\mu}_t(a_t^*)^\top \mathbf{x}_t(a_t^*)$ and $\mathbf{v}_t^* := \tilde{W}_t(a_t^*)^\top \mathbf{x}_t(a_t^*)$. By Corollary 2, with probability $1 - \delta$, we have that $T\mathbb{E}[r_t^* | H_{t-1}] \geq \text{OPT}$, and $\mathbb{E}[\mathbf{v}_t^* | H_{t-1}] \leq \frac{B}{T}\mathbf{1}$, where the expectation is over context $X_t$ given $H_{t-1}$. By the choice made by the algorithm,

$$\begin{aligned}
\tilde{r}_t - Z(\boldsymbol{\theta}_t \cdot \tilde{\mathbf{v}}_t) &\geq r_t^* - Z(\boldsymbol{\theta}_t \cdot \mathbf{v}_t^*) \\
\mathbb{E}[\tilde{r}_t - Z(\boldsymbol{\theta}_t \cdot \tilde{\mathbf{v}}_t) | H_{t-1}] &\geq \mathbb{E}[r_t^* | H_{t-1}] - Z(\boldsymbol{\theta}_t \cdot \mathbb{E}[\mathbf{v}_t^* | H_{t-1}]) \\
&\geq \frac{1}{T} \text{OPT} - Z\left(\boldsymbol{\theta}_t \cdot \frac{B}{T}\mathbf{1}\right).
\end{aligned}$$

Summing above inequality for $t = 1$ to $\tau$ gives the lemma statement. $\qquad\square$

**Lemma 9.**

$$\sum_{t=1}^{\tau} \boldsymbol{\theta}_t \cdot (\tilde{\mathbf{v}}_t - \frac{B}{T}\mathbf{1}) \geq B - \frac{\tau B}{T} - \mathcal{R}'(T).$$

*Proof.* Recall that $g_t(\boldsymbol{\theta}_t) = \boldsymbol{\theta}_t \cdot \left(\tilde{\mathbf{v}}_t - \frac{B}{T}\mathbf{1}\right)$, therefore the LHS in the required inequality is $\sum_{t=1}^{\tau} g_t(\boldsymbol{\theta}_t)$. Let $\boldsymbol{\theta}^* := \arg\max_{||\boldsymbol{\theta}||_1 \leq 1, \boldsymbol{\theta} \geq 0} \sum_{t=1}^{\tau} g_t(\boldsymbol{\theta})$. We use the regret definition for the OLalgorithm to get that $\sum_{t=1}^{\tau} g_t(\boldsymbol{\theta}_t) \geq \sum_{t=1}^{\tau} g_t(\boldsymbol{\theta}^*) - \mathcal{R}(T)$. Note that from the regret bound given in Lemma 4, $\mathcal{R}(T) \leq \mathcal{R}'(T)$.

**Case 1:** $\tau < T$. This means that $\sum_{t=1}^{\tau} (\mathbf{v}_t(a_t) \cdot \mathbf{e}_j) \geq B$ for some $j$. Then from (15) and (17), it must be that $\sum_{t=1}^{\tau} (\tilde{\mathbf{v}}_t \cdot \mathbf{e}_j) \geq B - \mathcal{R}'(T)$ so that $\sum_{t=1}^{\tau} g_t(\boldsymbol{\theta}^*) \geq \sum_{t=1}^{\tau} g_t(\mathbf{e}_j) \geq B - \frac{\tau B}{T} - \mathcal{R}'(T)$.

**Case 2:** $\tau = T$. In this case, $B - \frac{\tau}{T}B = 0 = \sum_{t=1}^{\tau} g_t(\mathbf{0}) \leq \sum_{t=1}^{\tau} g_t(\boldsymbol{\theta}^*)$, which completes the proof of the lemma. $\qquad\square$

Now, we are ready to prove Theorem 2, which states that Algorithm 1 achieves a regret of $Z\mathcal{R}'(T)$.
**Proof of Theorem 2.** Substituting the inequality from Lemma 9 in Lemma 8, we get

$$\sum_{t=1}^{\tau} \mathbb{E}[\tilde{r}_t | H_{t-1}] \geq \frac{\tau}{T} \text{OPT} + ZB\left(1 - \frac{\tau}{T}\right) - Z\mathcal{R}'(T)$$

Also, $Z \geq \frac{\text{OPT}}{B}$. Substituting in above,

$$\begin{aligned}
\mathbb{E}[\tilde{r}_{\text{sum}}] = \sum_{t=1}^{\tau} \mathbb{E}[\tilde{r}_t | H_{t-1}] &\geq \frac{\tau}{T} \text{OPT} + \text{OPT}(1 - \frac{\tau}{T}) - Z\mathcal{R}(T) \\
&\geq \text{OPT} - Z\mathcal{R}'(T)
\end{aligned}$$

From Steps 1 and 2, this implies a lower bound on $\mathbb{E}[\sum_{t=1}^{\tau} r_t(a_t)]$. The proof is now completed by using Azuma-Hoeffding to bound the actual total reward with high probability. $\qquad\square$

# F    Appendix for Section 3.3

**Proof of Lemma 5.** Let us define an "intermediate sample optimal" as:

$$\overline{\text{OPT}}^{\gamma} \quad := \quad \begin{array}{c} \max_q \quad \frac{T}{T_0}\sum_{i=1}^{T_0} \boldsymbol{\mu}_*^\top X_i \pi(X_i)]) \\ \text{such that} \quad \frac{T}{T_0}\sum_{i=1}^{T_0} W_*^\top X_i \pi(X_i) \leq B + \gamma \end{array} \tag{19}$$

Above sample optimal knows the parameters $\boldsymbol{\mu}_*, W_*$, the error comes only from approximating the expected value over context distribution by average over the observed contexts. We do not actually compute $\overline{\text{OPT}}^{\gamma}$, but will use it for the convenience of proof exposition. The proof involves two steps.

Step 1: Bound $|\overline{\text{OPT}}^{\gamma} - \text{OPT}|$.

Step 2: Bound $|\hat{\text{OPT}}^{2\gamma} - \overline{\text{OPT}}^{\gamma}|$

**Step 1** bound can be borrowed from the work on Online Stochastic Convex Programming in [4]: since $\boldsymbol{\mu}_*, W^*$ is known, so there is effectively full information before making the decision, i.e., consider the vectors $[\boldsymbol{\mu}_*^\top \mathbf{x}_t(a), W_*^\top \mathbf{x}_t(a)]$ as outcome vectors which can be observed for all arms $a$ *before* choosing the distribution over arms to be played at time $t$, therefore, the setting in [4] applies. In fact, $\hat{\text{OPT}}^{\gamma}$ as defined by Equation (F.10) in [4] when $A_t = \{[\boldsymbol{\mu}_*^\top \mathbf{x}_t(a), W_*^\top \mathbf{x}_t(a)], a \in [K]\}$, $f$ identity, and $S = \{\mathbf{v}_{-1} \leq \frac{B}{T}\}$, is same as $\frac{1}{T}$ times $\overline{\text{OPT}}^{\gamma}$ defined here. And using Lemma F.4 and Lemma F.6 in [4] (using $L = 1, Z^* = \text{OPT}/B$), we obtain that for any $\gamma \geq \left(\frac{T}{T_0}\right) 2m\sqrt{T_0 \log(T_0) \log(T_0 d/\delta)}$, with probability $1 - O(\delta)$,

$$\text{OPT} - \gamma \leq \overline{\text{OPT}}^{\gamma} \leq \text{OPT} + 2\gamma(\frac{\text{OPT}}{B} + 1). \tag{20}$$

For **Step 2**, we show that with probability $1 - \delta$, for *all* $\pi$, $\gamma \geq \left(\frac{T}{T_0}\right) 2m\sqrt{T_0 \log(T_0) \log(T_0 d/\delta)}$

$$|\sum_{i=1}^{T_0}(\hat{\boldsymbol{\mu}}_i - \boldsymbol{\mu}_*)^\top X_i \pi(X_i)| \leq \gamma \tag{21}$$

$$\|\frac{T}{T_0}\sum_{i=1}^{T_0}(\hat{W}_i - W_*)^\top X_i \pi(X_i)\|_\infty \leq \gamma \tag{22}$$

This is sufficient to prove both lower and upper bound on $\hat{\text{OPT}}^{2\gamma}$ for $\gamma \geq \left(\frac{T}{T_0}\right) 2m\sqrt{T_0 \log(T_0) \log(T_0 d/\delta)}$. For lower bound, we can simply use (22) for optimal policy for $\overline{\text{OPT}}^{\gamma}$, denoted by $\bar{\pi}$. This implies that (because of relaxation of distance constraint by $\gamma$) $\bar{\pi}$ is a feasible primal solution for $\hat{\text{OPT}}^{2\gamma}$, and therefore using (20) and (21),

$$\hat{\text{OPT}}^{2\gamma} + \gamma \geq \overline{\text{OPT}}^{\gamma} \geq \text{OPT} - \gamma.$$

For the upper bound, we can use (22) for the optimal policy $\hat{\pi}$ for $\hat{\text{OPT}}^{2\gamma}$. Then, using (20) and (21),

$$\hat{\text{OPT}}^{2\gamma} \leq \overline{\text{OPT}}^{3\gamma} + \gamma \leq \text{OPT} + 6\gamma(\frac{\text{OPT}}{B} + 1) + \gamma.$$

Combining, this proves the desired lemma statement:

$$\text{OPT} - 2\gamma \leq \hat{\text{OPT}}^{2\gamma} \leq \text{OPT} + 7\gamma(\frac{\text{OPT}}{B} + 1) \tag{23}$$

What remains is to proof the claim in (21) and (22). We show the proof for (22), the proof for (21) is similar. Observe that for any $\pi$,

$$\begin{aligned} \|\sum_{t=1}^{T_0}(\hat{W}_t - W_*)^\top X_t \pi(X_t)\|_\infty &\leq \sum_{t=1}^{T_0}\|(\hat{W}_t - W_*)^\top X_t \pi(X_t)\|_\infty \\ &\leq \sum_{t=1}^{T_0}\|\hat{W}_t - W_*\|_{M_t}\|X_t \pi(X_t)\|_{M_t^{-1}} \end{aligned}$$

where $\|\hat{W}_t - W_*\|_{M_t} = \max_j \|\hat{\mathbf{w}}_{tj} - \mathbf{w}_{*j}\|_{M_t}$.

Now, applying Lemma 2 to every column $\hat{\mathbf{w}}_{tj}$ of $\hat{W}_t$, we have that with probability $1 - \delta$ for all $t$,

$$\|\hat{W}_t - W_*\|_{M_t} \leq 2\sqrt{m\log(td/\delta)} \leq 2\sqrt{m\log(T_0 d/\delta)}$$

And, by choice of $p_t$

$$\|X_t \pi(X_t)\|_{M_t^{-1}} \leq \|X_t p_t\|_{M_t^{-1}}.$$

Also, by Lemma 3,

$$\sum_{t=1}^{T_0} \|X_t p_t\|_{M_t^{-1}} \leq \sqrt{mT_0 \ln(T_0)}$$

Therefore, substituting,

$$
\begin{aligned}
\|\sum_{t=1}^{T_0}(\hat{W}_t - W_*)^\top X_t \pi(X_t)\|_\infty &\leq (2\sqrt{m\log(T_0 d/\delta)}) \sum_{t=1}^{T_0} \|X_t p_t\|_{M_t^{-1}} \\
&\leq (2\sqrt{m\log(T_0 d/\delta)})\sqrt{mT_0 \ln(T_0)} \\
&\leq \frac{T_0}{T}\gamma
\end{aligned}
$$

$\square$