[Reviews · NeurIPS 2016]

Reviewer 1

Summary

The paper introduces a version of the contextual linear bandit problem with an additional vector-valued consumption process, along with the reward process, and a budget constraint added. The authors develop a bandit algorithm for the problem and derive a corresponding regret bound for it. The algorithm involves several ingredients like confidence ellipsoids from linear bandits, an online linear optimization subroutine using Mirror descent, and linear bandit exploration. The regret bound is orderwise seen to be tight in terms of the context dimension and time horizon problem parameters.

Qualitative Assessment

EDIT: I read the author response. The paper studies an interesting and relevant problem in online learning with contextual side information. It is quite smooth to read and does a good job at explaining the key ideas in the problem setting, where the problem fits into the literature on contextual/budgeted bandits, and how the algorithmic techniques and results compare to those of existing approaches. While the results in the paper do seem solid, I think some aspects of the presentation are lacking in clarity and could be significantly improved. * (line 183) What exactly is meant by the "mirror descent algorithm"? It seems that the authors use it as a black-box term for a low-regret algorithm for the problem of online linear optimization over the simplex, which could perhaps be something like the exponentially weighted update algorithm. The authors could perhaps be more explicit about this, and mention the exact link function being used in the mirror descent algorithm (e.g., entropy on the simplex). * I could not clearly understand the role of theta_t (which is optimized somehow by the OCO/mirror descent subroutine) in the overall algorithm, from the explanations given in the paper. Lines 207-209 promise a clear explanation of this sequence but it does not appear clearly later on, except for within the technical proofs. From the structure of the OCO algorithm, {theta_t} seems to be some kind of a Lagrange multiplier or dual variable sequence corresponding to the consumption < budget constraint, but it is still unclear to me at a high level why an update rule is required.

Confidence in this Review

1-Less confident (might not have understood significant parts)


Reviewer 2

Summary

This paper proposes a contextual bandit algorithm with knapsacks where the rewards and costs are linear functions of the feature vector, which is associated with each action and changes over time. The authors bound the regret of the proposed algorithm and compare it to the existing work. The paper is well written. The design of the algorithm could be motivated and explained better. The paper would be stronger if the authors evaluated their algorithm empirically.

Qualitative Assessment

*** Technical quality *** The paper is technical. My main comment is that the design of the proposed algorithm is not explained and justified well. In particular, the algorithm combines the ideas from both stochastic learning, such as the confidence intervals on \mu and W, and adversarial learning, the selection of \theta. At first, this raises eyebrows but I think that I understand. What I do not understand is why the learning of \theta is an adversarial problem. Can you elaborate on this? Since the learning problem is stochastic, I would think that it is possible to derive good \theta_t from the optimistic estimates of \mu and W, as in (4). *** Novelty / originality *** This paper proposes a new contextual bandit algorithm with knapsacks. The authors clearly argue that their regret bounds are better by instantiating more general results in their setting, such as contextual bandits with context-dependent policies. *** Potential impact or usefulness *** The impact of this paper is reduced because this is not the first paper on contextual bandits with knapsacks. Although the proposed algorithm achieves lower regret in theory, it may less practical because it requires an additional training period for estimating Z. Moreover, linear generalization is rarely good in practice and it is not clear what the sensitivity of the algorithm is to imperfect generalization. This can be typically only established by numerical experiments. I suggest that the authors evaluate their algorithm empirically and compare it to other bandit algorithms with knapsacks, both with context and without. *** Clarity and presentation *** The overall idea of this paper is clear and it is relatively easy to read. My detailed comments are below: Length: The main paper is 10 pages long. Therefore, it violates submission guidelines. Theorem 1 and line 94: I assume that both "log" and "ln" denote natural logarithms. Choose one and stick to it. Line 213: \tilde{\mu} should be \mu_\ast. Assumption 1: Say that the role of Z is to adjust costs such that they are comparable to rewards. Conclusions: Surprisingly none.

Confidence in this Review

2-Confident (read it all; understood it all reasonably well)


Reviewer 3

Summary

********************************************************* POST REBUTTAL: Thanks for the corrections. I now vote for acceptance. I hope the authors will spend the effort to improve the readability. ********************************************************* A new linear contextual bandit setting is proposed that includes the addition of a d-dimensional budget. This generalises existing budgeted bandits to the linear case. Everything is assumed to be i.i.d., and the budget usage of an action is assumed to be a linear function of the context for that action and some unknown parameter. Besides the setting (which is quite natural) the main contribution is a new algorithm and analysis showing that ignoring logarithmic factors enjoys a regret of O((1+OPT/B) m Sqrt(T)) where m is the dimension of the action space, B is the budget and OPT is the payoff of the optimal static strategy. Although there are no lower bounds, by combining existing lower bounds the authors argue that this is not much improvable, which is quite convincing.

Qualitative Assessment

The algorithm operates mostly in a quite standard way. Constructing confidence ellipsoids around the unknown parameters and acting optimistically subject to a budget penalty. The latter part is the most interesting because one has to try and learn the direction in the budget space for which the algorithm is most constrained, for which the authors use online mirror descent. As far as I know this is a completely novel idea. I found a lot to like in this paper. The setting and algorithm are interesting and the analysis is mostly quite good. There were also some weaknesses, however. For of all, there is a lot of sloppy writing, typos and undefined notation. See the long list of minor comments below. A larger concern is that some parts of the proof I could not understand, despite trying quite hard. The authors should focus their response to this review on these technical concerns, which I mark with ** in the minor comments below. Hopefully I am missing something silly. One also has to wonder about the practicality of such algorithms. The main algorithm relies on an estimate of the payoff for the optimal policy, which can be learnt with sufficient precision in a "short" initialisation period. Some synthetic experiments might shed some light on how long the horizon needs to be before any real learning occurs. A final note. The paper is over length. Up to the two pages of references it is 10 pages, but only 9 are allowed. The appendix should have been submitted as supplementary material and the reference list cut down. Despite the weaknesses I am quite positive about this paper, although it could certainly use quite a lot of polishing. I will raise my score once the ** points are addressed in the rebuttal. Minor comments: * L75. Maybe say that pi is a function from R^m \to \Delta^{K+1} * In (2) you have X pi(X), but the dimensions do not match because you dropped the no-op action. Why not just assume the 1st column of X_t is always 0? * L177: "(OCO )" -> "(OCO)" and similar things elsewhere * L176: You might want to mention that the learner observes the whole concave function (full information setting) * L223: I would prefer to see a constant here. What does the O(.) really mean here? * L240 and L428: "is sufficient" for what? I guess you want to write that the sum of the "optimistic" hoped for rewards is close to the expected actual rewards. * L384: Could mention that you mean |Y_t - Y_{t-1}| \leq c_t almost surely. ** L431: \mu_t should be \tilde \mu_t, yes? * The algorithm only stops /after/ it has exhausted its budget. Don't you need to stop just before? (the regret is only trivially affected, so this isn't too important). * L213: \tilde \mu is undefined. I guess you mean \tilde \mu_t, but that is also not defined except in Corollary 1, where it just given as some point in the confidence ellipsoid in round t. The result holds for all points in the ellipsoid uniformly with time, so maybe just write that, or at least clarify somehow. ** L435: I do not see how this follows from Corollary 2 (I guess you meant part 1, please say so). So first of all mu_t(a_t) is not defined. Did you mean tilde mu_t(a_t)? But still I don't understand. pi^*(X_t) is (possibly random) optimal static strategy while \tilde \mu_t(a_t) is the optimistic mu for action a_t, which may not be optimistic for pi^*(X_t)? I have similar concerns about the claim on the use of budget as well. * L434: The \hat v^*_t seems like strange notation. Elsewhere the \hat is used for empirical estimates (as is standard), but here it refers to something else. * L178: Why not say what Omega is here. Also, OMD is a whole family of algorithms. It might be nice to be more explicit. What link function? Which theorem in [32] are you referring to for this regret guarantee? * L200: "for every arm a" implies there is a single optimistic parameter, but of course it depends on a ** L303: Why not choose T_0 = m Sqrt(T)? Then the condition becomes B > Sqrt(m) T^(3/4), which improves slightly on what you give. * It would be nice to have more interpretation of theta (I hope I got it right), since this is the most novel component of the proof/algorithm.

Confidence in this Review

2-Confident (read it all; understood it all reasonably well)


Reviewer 4

Summary

The authors provide an algorithm for solving the linear contextual bandit with knapsacks problem. In this problem, the algorithm plays a series of rounds where in each round the algorithm is presented with a "context vector" x \in R^m and must choose an action in [1..K], or a "no-op" action. The algorithm then receives a reward r and a resource consumption vector v. r=0 and v=0 for the no-op action. r and v are stochastic values whose expectation for each action is a fixed linear function of the context. The goal is to maximize the sum of rewards while keeping each component of the sum of consumption vectors below some budget B. The authors provide an algorithm that achieves with high probability achieves regret \tilde O( optimal_loss/B m sqrt(T) ).

Qualitative Assessment

This paper is clearly written and provides an answer to an interesting variation on contextual bandits. It would have been valuable to see some experimental validation that the algorithm performs as stated. The linear contextual bandits with knapsacks problem is sufficiently narrow that the algorithm will probably not see widespread use, although the advertising case is potentially valuable - again, some experiments are necessary to make this convincing.

Confidence in this Review

2-Confident (read it all; understood it all reasonably well)


Reviewer 5

Summary

In this paper the authors are considering linear contextual bandits, when there are constraints on resource consumption. The authors are combining techniques from previous well known techniques and tackle some of the issues that don't happen in the unstructured version of the problem.

Qualitative Assessment

I liked the idea that the authors have set out to address. I can see that the authors put in hard work in putting this paper together and I commend them for their efforts. However, I do believe that this paper might require the addition of a few minor additions. I would suggest the following: The last paragraph of the introduction should introduce what the rest of the paper holds, a gist of the different sections of the rest of the paper essentially. I felt that the paper ends rather abruptly, so it might be really helpful to include a conclusions section before the references. I praise and appreciate that the authors used gender positive language such as "in every round ... she needs to" etc. I appreciate again the novel idea that the authors are trying to tackle. I wish them best of luck.

Confidence in this Review

1-Less confident (might not have understood significant parts)


Reviewer 6

Summary

This paper considers linear contextual bandit problem with budgeted resources (knapsack problem). Different from previous works, this paper assumes linearity on both reward and consumption. Based on such assumption, authors utilize the state-of-the-art techniques from linear contextual bandits, bandits with knapsacks and OSPP, prove that with high probability, algorithm proposed in this paper (linCBwK) could achieve O(m \sqrt{\log(T) T}) regret bound with budget needs of \Omega{mT^{3/4}}.

Qualitative Assessment

The problem is motivated by an novel assumption of linear reward and consumption generation. The paper is well written and pleasant to read. Authors prove that their proposed algorithm could achieve a near optimal regret bound. The sketch of the proof is clear and seems solid to me (however I did not check every proof in detail). Although the regret bound of the prosoed algorithm is sound and promising, authors also mentioned that requirment of budget is \Omega{mT^{3/4}}, which ideally could be \Omega{mT^{1/2}}. The skech of proof is mostly following [5], and it would be better if authors could provide more intuition and analysis regarding why this budget requirement is inevitable under such framework. *** more intuition and comparison with [5] is provided in rebuttal. ***

Confidence in this Review

2-Confident (read it all; understood it all reasonably well)